# Immunological and Haematological Relevance of Helminths and *Mycobacterium tuberculosis* Complex Coinfection among Newly Diagnosed Pulmonary Tuberculosis Patients from Bobo-Dioulasso, Burkina Faso

**DOI:** 10.3390/biomedicines12071472

**Published:** 2024-07-03

**Authors:** Diakourga Arthur Djibougou, Gloria Ivy Mensah, Achille Kaboré, Inoussa Toé, Leon Tinnoga Sawadogo, Palpouguini Felix Lompo, Amariane M. M. Kone, Hervé Hien, Clement Ziemlé Meda, Adjima Combary, Bassirou Bonfoh, Kennedy Kwasi Addo, Adrien Marie-Gaston Belem, Roch Konbobr Dabiré, Jonathan Hoffmann, Matthieu Perreau, Potiandi Serge Diagbouga

**Affiliations:** 1Doctoral School of Natural Sciences and Agronomy, Université Nazi BONI, Bobo-Dioulasso 1091, Burkina Faso; inoussatoe11@gmail.com (I.T.); sawtileon@yahoo.fr (L.T.S.); medacle1@yahoo.fr (C.Z.M.); belemamg@hotmail.fr (A.M.-G.B.); 2Infectious Diseases Program, Centre MURAZ, Institut National de Santé Publique, Bobo-Dioulasso 1091, Burkina Faso; koneama@yahoo.fr (A.M.M.K.); hien_herve@hotmail.com (H.H.); dabireroch@gmail.com (R.K.D.); 3Institut de Recherche en Sciences de la Santé, CNRST, Bobo-Dioulasso 545, Burkina Faso; 4Department of Bacteriology, Noguchi Memorial Institute for Medical Research, College of Health Sciences, University of Ghana, Legon, Accra 00233, Ghana; gmensah@noguchi.ug.edu.gh (G.I.M.); kaddo@noguchi.ug.edu.gh (K.K.A.); 5Infectious Disease and Health Systems (IDHS), FHI 360, Washington, DC 20037, USA; kachille@fhi360.org; 6National Tuberculosis Programme, Ministry of Health and Public Hygiene, Ouagadougou 01 P.O. Box 690, Burkina Faso; adjicomb@yahoo.fr; 7Etudes Formation et Recherches Développement en Santé (EFORDS), Ouagadougou 10 P.O. Box 13064, Burkina Faso; felixlompo@gmail.com; 8Centre Suisse de Recherches Scientifique de Côte d’Ivoire, Adiopodoumé 01 P.O. Box 1303, Côte d’Ivoire; bassirou.bonfoh@csrs.ci; 9Département Médical et Scientifique, Fondation Mérieux, 17 rue Bourgelat, 69002 Lyon, France; jonathan.hoffmann@fondation-merieux.org; 10Faculty of Biology and Medicine, Université de Lausanne, 1010 Lausanne, Switzerland; matthieu.perreau@chuv.ch

**Keywords:** *M. tuberculosis* complex and helminth coinfection, CD4 T cells, CD4:CD8 ratio, immune–haematological parameters

## Abstract

The effect of helminthiasis on host immunity is a neglected area of research, particularly in tuberculosis (TB) infection. This study aimed to evaluate the effect of helminthiasis on immunological and haematological parameters in newly diagnosed TB patients in Bobo-Dioulasso. After all biological analyses, we formed three subpopulations: group 1 (*n* = 82), as control, were participants without helminthic or *Mycobacterium tuberculosis* complex infection (*Mtb*−/Helm−), group 2 (*n* = 73) were TB patients without helminthic infection (*Mtb*+/Helm−), and group 3 (*n* = 22) were TB patients with helminthic infection (*Mtb*+/Helm+). The proportion of helminth coinfection was 23.16% (22/95) in TB patients, and *Schistosoma mansoni* infection was found in 77.3% (17/22) cases of helminthiasis observed in this study. A low CD4 T cell count and a low CD4:CD8 ratio were significantly associated with concomitant infection with helminths and the *Mtb* complex (*Mtb*+/Helm+) compared to the other groups (*p* < 0.05). However, there was no statistically significant difference in the CD8 median among the three participating groups (*p* > 0.05). Lymphopenia, monocytosis, thrombocytosis, and hypochromic microcytic anaemia were the haematological defects observed in the *Mtb+*/Helm+ and *Mtb+*/Helm− patients. Exploring these types of immune–haematological biomarkers would be a valuable aid in diagnosing and a better follow-up and monitoring of the tuberculosis–helminthiasis coinfection.

## 1. Introduction

Helminths are a group of metazoan parasites that infect over 2 billion people worldwide [1]. They consist of nematodes, trematodes, and cestodes and remain the most common parasitic infection agents worldwide [2]. Similarly, tuberculosis (TB) affects more than one-third of the world’s population and remains a major public health problem, with 1.4 million deaths and more than 10 million new cases per year [3,4].

The geographical distributions of helminth infections and TB overlap considerably, especially in developing countries, which increases the risk of coinfection with both diseases [5,6,7,8]. This scenario leads to speculations that helminths may reactivate or aggravate TB. Indeed, in case of coinfection, helminths induce immunomodulatory reactions through a humoral T helper-2 (Th2) immune response, characterised by the production of cytokines, such as IL-4, IL-5, IL-9, IL-10, and IL-13, and high levels of immunoglobulin E (IgE), eosinophils, mast cells, and regulatory T cells (Tregs) [9,10], at the expense of the host’s protective T helper-1 (Th1) cell-mediated immune response against the Mtb complex agents [11,12,13]. Deep and prolonged suppression of Mtb-specific CD4, CD8 T cell responses correlate with low production of T helper 1 (Th1) cytokines, such as interferon (IFN)-γ, a good biomarker of TB infection or active TB disease [14]. In Ethiopia, a significant reduction in IFN-γ-producing CD4 and CD8 T cells has been noted in patients with active pulmonary TB co-infected with Schistosoma mansoni [15]. In contrast, others suggest that chronic infection has no immunological impact on the exacerbation of Mtb infection. In addition, helminthic infection and tuberculosis cause profound haematological abnormalities such as anaemia, lymphopenia, eosinophilia, and monocytosis [16,17]. Exploring these parameters would therefore be a valuable aid in diagnosing and monitoring of TB patients.

In Burkina Faso, the frequency of parasitosis and the incidence of TB remain high and therefore constitute a public health problem. The prevalence of intestinal parasitic infections at the reference hospital in Bobo-Dioulasso was 65.3% in 2015 [18], and the incidence of TB is estimated at 45/100,000 inhabitants in 2021 [4]. Yet, to our knowledge, data on the helminth infection’s effect on T cell subpopulations’ response and the variation of haematological parameters during TB are lacking. Therefore, in this study, we analysed the influence of helminths on immuno-haematological parameters in patients with newly diagnosed active pulmonary tuberculosis, with or without concomitant helminthiasis, in the city of Bobo-Dioulasso in western Burkina Faso.

## 2. Materials and Methods

### 2.1. Ethical Statement

The study protocol was approved by the Burkina Faso Health Research Ethics Committee (Ref. 2017-07-106/CERS). Authorizations for data collection were issued by the Ministry of Health and the National Tuberculosis Programme. All participants gave written informed consent after being explained this study’s procedures, risks, and benefits. Participants infected with parasites were treated free of charge according to their parasitic infestation with the following drugs: Albendazole (400 mg), Praziquantel (600 mg), and metronidazole (500 mg).

### 2.2. Type of Study, Area and Period

This comparative cross-sectional study was conducted from 2 March 2019 to 30 July 2021 in Bobo-Dioulasso (11°10′42″ N; 4°17′35″ W), the economic capital in the western part of the country. Indeed, this study was launched on 2 March 2019 after getting clearance from the ethical committee, all authorizations from the Ministry of Health, as well as the National Tuberculosis Program in August 2018, and the research team began to enrol the study participants. The field work was executed from 2 March 2019 to 30 June 2021, and the lab analyses from 2 March 2019 to 30 July 2021. Despite the introduction of mass drug distribution in 2001, parasitosis is still prevalent. The overall prevalence of intestinal parasitic infections at the reference hospital in Bobo-Dioulasso was 65.3% in 2015 [18], and the incidence of TB was estimated at 45/100,000 inhabitants in 2021 (WHO, 2022). The collection sites for this study were the regional TB control centre (CRLAT), the Dafra medical centre, the Do medical centre, and the abattoir of Bobo-Dioulasso (Figure 1).

### 2.3. Populations and Sample Collection

The participants in this study consisted of newly diagnosed pulmonary TB patients from the Regional TB Control Centre of Bobo-Dioulasso (CRLAT) and healthy control subjects from the Medical Center of Dô (CMA de Dô) and the Medical Center of Dafra (CMA de Dafra) (Figure 1), as well as participants free of symptoms and signs suggestive of TB, who gave informed consent. Participants with a positive HIV serology, as well as healthy participants with helminth infection were not included in this study. In the present study, participants were classified into three subgroups: (1) those without helminthic infection (*Mtb*-/Helm−), (2) TB patients without helminthic infection (*Mtb*+/Helm−), and (3) TB patients with helminthic infection (*Mtb*+/Helm+) (Figure 2).

### 2.4. Laboratory Procedures

A stool sample, a urine sample (for parasitological analysis), and a 4 mL blood sample in an ethylenediaminetetraacetic acid tube (for immuno-haematological tests) were collected from each participant. All samples were sent to the Centre MURAZ centre with the completed collection sheets for analysis.

#### 2.4.1. Parasitological Analysis

Treatment of stool samples:

The formalin–ether and Kato–Katz concentration technique: Stool samples were processed using the formalin–ether concentration method described previously [19]. The concentration pellets were placed on slides and examined using 10× and 40× objectives. In addition, all samples were treated with Kato–Katz for helminth eggs, as previously described [20].

Processing of urine samples: urine samples were examined qualitatively for *Schistosoma* spp. using the urine sedimentation method and the rapid circulating cathodic antigen test (CCA-POC), as described previously [21].

Serological diagnosis of lymphatic filariasis and HIV: Antibodies to lymphatic filarial IgM and IgG testing was performed using the Filariasis IgG/IgM Combo rapid test (CTK Biotech, Poway, CA, USA) on serum from the study participants following the manufacturer’s procedures.

HIV testing was performed using the Alere Determine HIV rapid test (Alere; San Diego, CA, USA) and the OnSite HIV 1/2 Ab Plus Combo Rapid test (CTK Biotech, USA) in accordance with the Burkina Faso HIV testing algorithm.

#### 2.4.2. Whole Blood Immuno–Haematological Analyses

The complete blood count (CBC) was performed using the Sysmex XN-550 (Sysmex Europa; Bornbarch, Germany), within 6 h after collection, while following the manufacturer’s procedures.

CD4, CD8 T cells counts and CD4:CD8 ratio were performed on the BD FACSCount™ (Becton Dickinson Biosciences, San Jose, CA, USA). The reagent kit is provided in a two-tube format containing the antibodies tube of CD4/CD3 reagents with reference beads and a tube of CD8/CD3 reagents with reference beads. An amount of 50 μL of EDTA uncoagulated whole blood was added to the two tubes (CD4/CD3 reagent tube, CD8/CD3 reagent tube) using a preprogrammed electronic pipette. They were vortexed for 5 s and incubated in the dark at room temperature for 30 min. Then, 50 μL of a fixative solution was added to the tubes. The tubes were vortexed, and the non-lysed stained sample was analysed in FACSCount™ using the standard software.

### 2.5. Statistical Analysis

Data were entered into Microsoft Excel and then exported to Stata version 14 IC (Stata Corp, College Station, TX, USA) for statistical analysis. Descriptive statistics were performed using proportions for categorical variables and mean or median for continuous variables. In addition, to assess the relationship between immunological and haematological parameters and the status of participants defined as *Mtb*−/Helm−, *Mtb*+/Helm−, and *Mtb*+/Helm+, we used ANOVA or the Kruskal Wallis test. Then, we performed a pairwise comparison between these results using the Student’s *t*-test or the Wilcoxon test, while correcting for multiple testing.

## 3. Results

### 3.1. Characteristics of Study Subjects

The objective of the present study was to evaluate the mean level of lymphocyte subpopulations and the variation of haematological parameters in pulmonary TB patients with concomitant (*n* = 22) or without (*n* = 73) concomitant infection with helminths, compared to the control group (supposedly healthy individuals) (*n* = 82) (Figure 2). Based on the available data, we divided the study population into three groups, whose demographic profiles are summarised in Table 1. The median age was 37 (iqr = 17), 34 (iqr = 17), and 34.5 (iqr = 19) for *Mtb*−/Helm−, *Mtb*+/Helm, and *Mtb*+/Helm+, respectively.

### 3.2. Profile of Helminths Encountered in the Population with TB

TB–Helminthiasis coinfection was reported in 23.16% (22 of 95) of all patients with TB, and they were positive for at least one helminth species. Interestingly, *Schistosoma* spp. was the most identified in this study in 77.3% (17/22) cases of helminthiasis (Table 2).

### 3.3. Effect of Helminth Infection on Immuno-Haematological Parameters

#### 3.3.1. Analysis of the Relationship between Lymphocyte Subpopulations (CD4, CD8), CD4:CD8 Ratio, and Helminth Infection Status

In this study, the mean CD4 T cell counts is significantly lower in both TB patients’ subgroups (*Mtb*+/Helm+ and *Mtb*+/Helm−) compared to the control group (*p* < 0.001 for both groups; Figure 3A). In contrast, there was no statistically significant difference in the median CD8 cell count between the control group, *Mtb*+/Helm−, and *Mtb+*/Helm+ individuals (*p* = 0.64 and *p* = 0.57, respectively) nor between *Mtb*+/Helm− and *Mtb*+/Helm+ individuals (*p* = 0.76) (Figure 3B). When considering the ratio, the same trend as for CD4 lymphocytes was observed (Figure 3C); indeed, coinfection (*Mtb*+/Helm+) resulted in a further low ratio compared to healthy *Mtb*−/Helm− individuals (*p* = 0.00006). The same was true for the *Mtb*+/Helm− patients, compared to the *Mtb*+ coinfected patients (*p* = 0.025).

#### 3.3.2. Relationship between Haematological Parameters, TB, and Helminth Infection Status

The mean values of the blood components platelets, red blood cells, and white blood cells, in addition to their subsets, including neutrophils, eosinophils, basophils, lymphocytes, monocytes, and haematimetric constants, were analysed and compared between the subgroups (Figure 4A–M). The mean leukocyte values, although normal in the *Mtb*+/Helm− and *Mtb*+/Helm+ patients, tended to increase compared to controls (*p* < 0.05 for both groups). Still, the presence or absence of helminth coinfection did not affect the leukocyte level (*p* = 0.15). In contrast, the mean level of lymphocytes was significantly reduced between the *Mtb*+/Helm− and the *Mtb*+/Helm+ patients, compared to the control subjects (*p* < 0.05 for both groups). However, total lymphocytes were significantly higher in *Mtb*−/Helm− than in *Mtb*+/Helm+ (*p* < 0.0001), and total lymphocytes were significantly higher in the *Mtb*+/Helm− group compared to the *Mtb*+/Helm+ group (*p* = 0.00084), suggesting lymphopenia in the face of TB–Helminthiasis coinfection. The number of monocytes was significantly higher in the *Mtb*+/Helm− and *Mtb*+/Helm+ patients compared to the control subjects (*p* < 0.05). However, we did not observe a statistically significant difference in the number of eosinophils between the *Mtb*+/Helm− and *Mtb*+/Helm+ patients and controls (*p* > 0.05) (Figure 4C). These results show that subjects with both active TB and concomitant helminthic infection are characterised by lymphopenia and monocytosis, without eosinophilia. For platelets, there was also notable thrombocytosis in the *Mtb*+/Helm+ and *Mtb*+/Helm− coinfected patients compared to the healthy *Mtb*−/Helm− subjects (*p* < 0.05) (Figure 4M). The mean haemoglobin rate in this study was significantly reduced in the *Mtb*+/Helm− and *Mtb*+/Helm+ patients compared to the control subjects (*p* < 0.001 for both groups). However, there was no statistically significant difference between the *Mtb*+/Helm− and *Mtb*+/Helm+ patients (*p* > 0.05) with respect to anaemia (Figure 4H). The same pattern was observed for MCV/VGM, MCHT/TCMH in both groups of TB patients, compared to the control group (Figure 4J,L). These data indicate that *Mtb*+/Helm− and *Mtb*+/Helm+ subjects have hypochromic microcytic anaemia, although no statistical difference is observed between these TB subjects.

## 4. Discussion

The impact of helminth infections on host susceptibility to *M. tuberculosis* complex agents is still poorly studied in West Africa. Therefore, our study explored the variations of immuno-haematological parameters in various contexts, such as in patients with microbiologically confirmed pulmonary TB, with or without concomitant helminthiasis, and healthy subjects.

We observed that 23.16% of the TB patients in our study were co-infected with helminthiasis. This frequency is similar to that reported in other studies in Arba Minch, Ethiopia, 24.4% [22], Tanzania, 31.8% [23], Brazil, 27.5% [17], and northwest Ethiopia, 29% [24]. However, our rate was lower than that reported in Ethiopia by Elias et al. [25] at 71%. The probable explanation would be the differences in the study period. Indeed, ours was conducted more recently, in a context where helminth control intervention strategies such as hygiene and sanitation approaches and mass drug distributions have significantly reduced the incidence of helminthiasis. These results were discussed in detail in our previous study [26]. In the co-infected patients (*Mtb*+/Helm+; *n* = 22), helminthiasis was mainly due to *Schistosoma mansoni*, 77.27% (17/22), followed by *Dicrocoelium dentriticum*, 13.63% (3/22), *Enterobius vermicularis*, 9.09% (2/22), and finally, *Wuchereria bancrofti* and *Hymnelopis nana*, 4.03% (1/22), in the TB patients. Similar results have been reported in other studies [14,27,28,29]. However, it is essential to understand the contribution of helminths in host immunological dysregulation, given that knowledge of the dynamics of immuno-haematological parameters is crucial for patients’ diagnostic and therapeutic decision support.

In the present study, we noted that the mean value of CD4 T cell count and CD4:CD8 ratio were significantly lower in *Mtb*+/Helm− and *Mtb*+/Helm+ patients compared to control participants (*p* < 0.05) (Figure 3A). In addition, the reduction of CD4 T cells in the blood of *Mtb*+/Helm+ patients compared to *Mtb*+/Helm− patients was statistically significant (*p* < 0.05), suggesting a negative impact of helminths. These results are similar to previous studies that reported a significant reduction in CD4 counts in TB patients with or without concomitant helminth infection in Brazil [17,30] and in Ethiopia [15]. In India, George et al. reported that coinfection with *Strongyloides stercoralis* was associated with a low CD4 + T cells co-expressing the Th1-like response (TNF-α/IFN-γ or IL-2/IFN-γ/TNF-α) in people with active TB [27]. This significant reduction in CD4 in *Mtb*+/Helm+ could be explained by tissue invasion of pathogens, followed by chronic CD4 sequestration at sites of infection or increased lymphocyte apoptosis during concomitant infection. In contrast, such results would be expected, given that helminth infections promote a Th2 immune response in coinfected individuals [13]. However, no statistically significant difference was observed in the mean CD8 T cell count between helminth-infected and uninfected patients (*p* > 0.05). This result is similar to that reported in Gondar, Ethiopia, by Wondmagegn et al. [31] in patients with the same characteristics. A significantly lower CD4 count and a relatively stable CD8 count in both subgroups resulted in a lower CD4/CD8 ratio in the coinfected patients in our study. These results are in line with that of Kalinkovich et al. [32], suggesting that a low CD4 T cell count and a high CD8 T cell count would be associated with chronic helminth infection [32,33,34]. This calls for better management, as it would indicate a weakened clinical picture due to suppression of the host immune system during coinfection [28]. However, this should be taken with caution because, in other contexts, helminthic infection would not impact host immunity in the same manner as chronic filarial infections in the mouse model [35].

The mean leukocyte values in *Mtb*+/Helm− and *Mtb*+/Helm+ patients were within normal limits but showed a tendency to be lower compared to controls (*p* < 0.05 for both groups). However, the presence or absence of helminth coinfection did not affect the leukocyte levels in TB patients (*p* = 0.15). These results are similar to those reported by Resende et al. in Brazil [17].

Monocytosis was observed in both *Mtb*+/Helm− and *Mtb*+/Helm+ patients compared to control subjects (*p* < 0.05). The same finding has been reported by others [17] and could be explained by the high probability of monocytosis in the face of chronic inflammation resulting from infections such as TB [36].

Furthermore, according to several authors, helminthiasis has been consistently associated with eosinophilia [12,24,33]. Contrary to what would be expected, helminths did not cause eosinophilia in our study. Similar results have been reported in Indonesia in a study population similar to ours [37]. One possible reason is that parasites induce transient eosinophilia during the tissue invasive stages of their development [38,39,40,41,42]. However, we have reservations, given the number of coinfected patients compared to the other study subgroup (*Mtb*+/Helm+: *n* = 22 only vs. *Mtb*+/Helm−: *n* = 73).

Lymphopenia was observed in all TB patients compared to control subjects. Similar results have been reported in Pakistan [43], in South Africa [44], Brazil [17], as well as elsewhere in the world [45]. Furthermore, this lymphopenia was statically significant in *Mtb*+/Helm+ co-infected individuals compared to single TB patients (*p* < 0.05). In Brazil, similar results have been reported in *Strongyloides stercoralis* co-infected patients compared to uncomplicated TB patients [17]. A recent study reported the correlation between lymphocyte depletion and TB treatment failure [45]; further studies should assess the impact of helminthiasis–tuberculosis coinfection on the efficacy of anti-TB treatment by measuring the kinetics of specific biomarkers before, during, and after treatment in our setting.

Our study reported thrombocytosis in all subgroups of TB patients, with no statistical difference between these subgroups (*Mtb*+/Helm− and *Mtb*+/Helm+). Observational studies have also reported thrombocytosis in TB patients [46,47,48]. This could be explained by the fact that platelets accelerate an innate immune response during the invasion of the pathogen in question [49]. Also, various cytokines, such as IL-6, involved in granuloma formation promote platelet production, which could be the basis for thrombocytosis. However, other studies have associated thrombocytopenia with TB in India [50].

Anaemia was observed in all TB patients as well, compared to control subjects. Similar findings have been reported in several studies [34,43,50]. However, there was no statistically significant difference in mean haemoglobin levels between the two subgroups (*Mtb*+/Helm− and *Mtb*+/Helm+). The anaemia was microcytic hypochromic. Baluku et al. [34] reported the same anaemia characteristics in a similar study population. The likely explanation for this is that both active TB and helminthiasis are associated with reduced haemoglobin levels [22].

Our study was limited by unequal sizes of the study groups, which introduced bias. In addition, a thorough characterisation of the immune cells to assess their functional status was not performed for logistical reasons.

## 5. Conclusions

Our study revealed that concomitant helminth and TB infection was associated with a low CD4, a low CD4/CD8 ratio and lymphopenia. In addition, monocytosis, thrombocytosis, and hypochromic microcytic anaemia were the haematological defects observed in *Mtb*+/Helm+ and *Mtb*+/Helm− patients. This study provided a preliminary understanding of the dynamics of the immuno-haematological parameters, which would help in the diagnostic and therapeutic decision-making for patients in West African settings. However, studies involving the in-depth exploration of the functionality of the immune system cells are needed to identify the best immunological biomarkers of exposure and/or predisposition to TB.

## Figures and Tables

**Figure 1 biomedicines-12-01472-f001:**
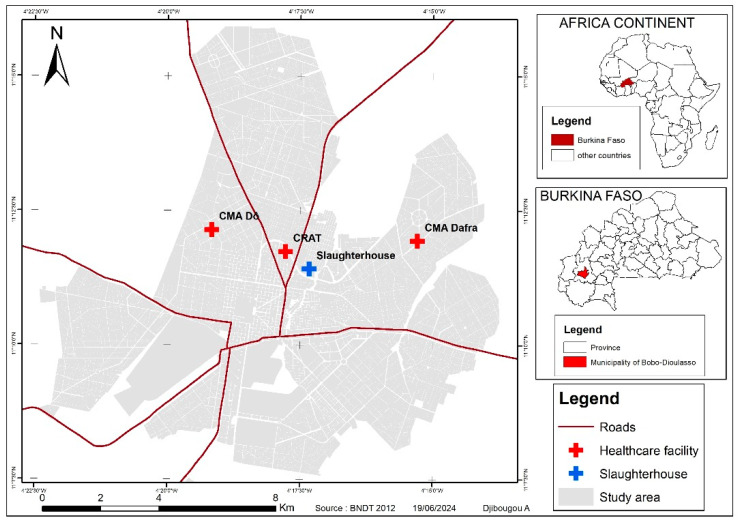
Map of the study area.

**Figure 2 biomedicines-12-01472-f002:**
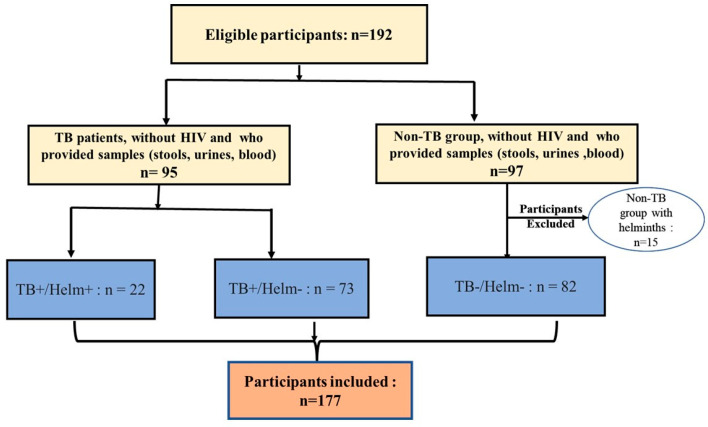
Study flowchart.

**Figure 3 biomedicines-12-01472-f003:**
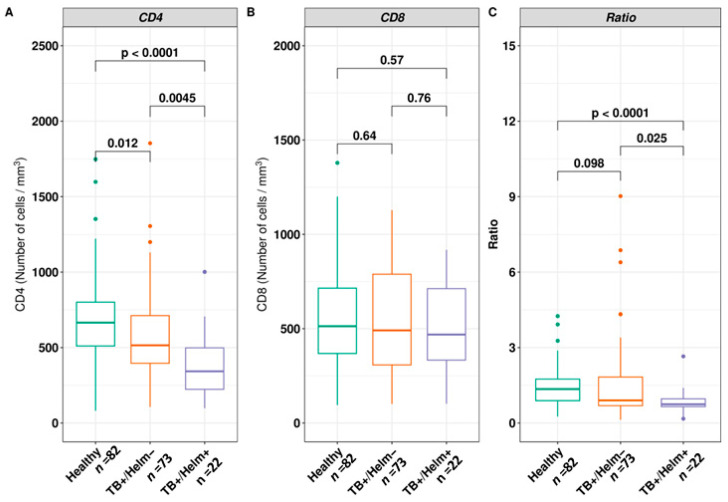
Relationship between immunological parameters and helminth infection status. (**A**–**C**) Mean of CD4, CD8 T cells subpopulations and CD4:CD8 ratio in blood from healthy participants (TB−/Helm− or *Mtb*−/Helm−), TB patients without helminths (TB+/Helm− or *Mtb*+/Helm−), and TB patients with helminth infection (TB + Helm or *Mtb*+/Helm+) patients. Results are expressed as mean or median. Significant differences between groups are depicted in the figure with their respective *p* -values and were calculated using appropriate tests.

**Figure 4 biomedicines-12-01472-f004:**
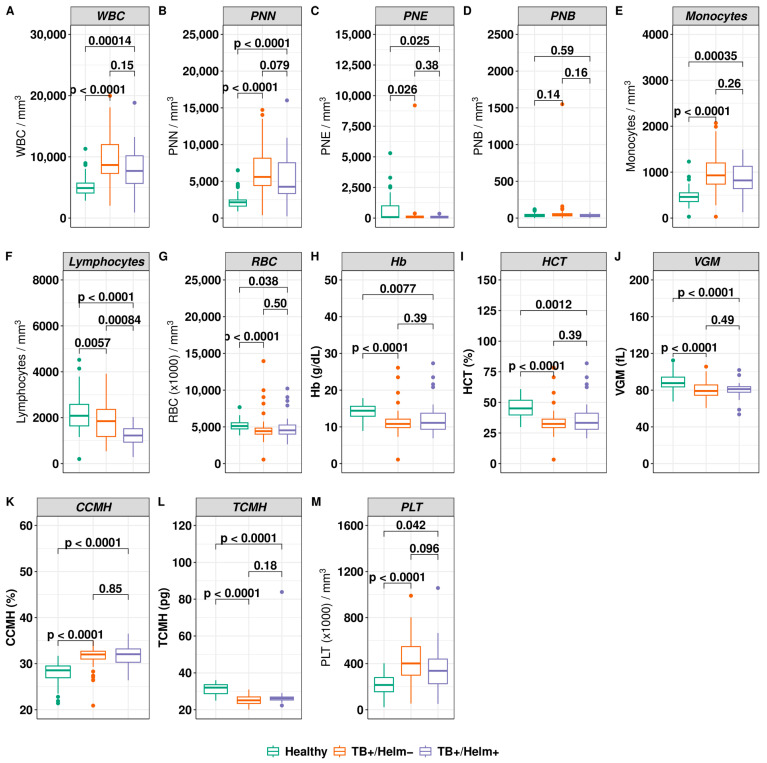
Relation between haematological parameters and helminth infection status. (**A**) Mean values of the white blood cells (WBC); (**B**) mean of WBC subsets such as neutrophils (**C**) eosinophils, (**D**) basophils, (**E**) monocytes, (**F**) lymphocytes; (**G**) mean of red blood cells (RBC) and means of haematimetric constants such as: (**H**) haemoglobin rate, (**I**) haematocrit rate, (**J**) VGM/MCV mean, (**K**) corpuscular haemoglobin concentration (MCHC/CCMH); (**L**) mean of MCH/TCMH; and (**M**) mean of platelets (PLT). % = percent, fL = femtoliter, g/dL = gramme per decilitre, Pg = picogramme. Significant differences between groups are depicted in the figure with their respective *p*-values and were calculated using appropriate tests.

**Table 1 biomedicines-12-01472-t001:** Description of age and sex of TB patients (*Mtb*+/Helm−), patients with both TB and helminths (*Mtb*+/Helm +), and controls (*Mtb*−/Helm−).

Variables	Category	Groups			*p*-Value
		*Mtb*−/Helm− *n* = 82	*Mtb*+/Helm− *n* = 73	*Mtb*+/Helm+ *n* = 22	
Age, median (IQR)		37 (17)	34 (17)	34.5 (19)	0.629 ^a^
Gender, *n* (%)	Male	58 (70.73)	62 (84.93)	19 (86.36)	0.063 ^b^
	Female	24 (29.97)	11 (15.07)	3 (13.64)	

^a^ *p* value for the median age using the Kruskall Wallis test; ^b^ *p* value for gender using the Fisher test.

**Table 2 biomedicines-12-01472-t002:** Profile of helminths in concomitant infection with *Mtb*, *n* = 22.

Helminth Species	TB Patients Coinfected *n* = 22	Frequency %
*Schistosoma mansoni*	17	77.27
*Dicrocoelium denditricum*	3	13.63
*Enterobius vermicularis*	2	9.09
*Wuchereria bancrofti*	1	4.54
*Hymnelopis nana*	1	4.54

## Data Availability

Data are contained within the article.

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
