# Peer review of "Immunological and Haematological Relevance of Helminths and Mycobacterium tuberculosis Complex Coinfection among Newly Diagnosed Pulmonary Tuberculosis Patients from Bobo-Dioulasso, Burkina Faso"

_biomedicines, 2024, doi:10.3390/biomedicines12071472_

Round 1

Reviewer 1 Report

Comments and Suggestions for Authors

The authors describe the relevance of worm and mycobacterial infections in newly diagnosed patients in Bobo-Dioulasso, Burkina Faso. The topic is relevant and innovative as little attention is paid to helminths in tuberculosis patients. An ethics vote was available.

A total of 192 patients were examined. 82 had neither tuberculosis nor worm infections, 73 had only tuberculosis and 22 had a co-infection. Patients with a co-infection in particular had significantly lower CD4 cell counts in their blood.

The work is logically structured. The introduction provides relevant background information. The framework conditions and methods of the study are described in a comprehensible manner. The results show only gradual differences.

A detailed analysis of the T cells would be desirable, but cannot be supplemented under the specific study conditions.

Author Response

Response to Reviewer 1 X Comments:

Best regards

Review Report Form

Open Review

(x) I would not like to sign my review report
( ) I would like to sign my review report Quality of English Language

(x) I am not qualified to assess the quality of English in this paper
( ) English very difficult to understand/incomprehensible
( ) Extensive editing of English language required
( ) Moderate editing of English language required
( ) Minor editing of English language required
( ) English language fine. No issues detected

Yes

Can be improved

Must be improved

Not applicable

Does the introduction provide sufficient background and include all relevant references?

(x)

( )

( )

( )

Is the research design appropriate?

(x)

( )

( )

( )

Are the methods adequately described?

(x)

( )

( )

( )

Are the results clearly presented?

(x)

( )

( )

( )

Are the conclusions supported by the results?

(x)

( )

( )

( )

Comments and Suggestions for Authors

The authors describe the relevance of worm and mycobacterial infections in newly diagnosed patients in Bobo-Dioulasso, Burkina Faso. The topic is relevant and innovative as little attention is paid to helminths in tuberculosis patients. An ethics vote was available.

A total of 192 patients were examined. 82 had neither tuberculosis nor worm infections, 73 had only tuberculosis and 22 had a co-infection. Patients with a co-infection in particular had significantly lower CD4 cell counts in their blood.

The work is logically structured. The introduction provides relevant background information. The framework conditions and methods of the study are described in a comprehensible manner. The results show only gradual differences.

Answer : We thank the reviewer for his critical reading and appreciations.

A detailed analysis of the T cells would be desirable, but cannot be supplemented under the specific study conditions.

Answer: We have briefly described the procedures as follows : ”CD4, CD8 T cells counts and CD4:CD8 ratio were performed on the BD FACSCount™ (Becton Dickinson Biosciences, San Jose, California, USA). The reagent kit is provided in a two-tube format containing the antibodies tube of CD4/CD3 reagents with reference beads and tube of CD8/CD3 reagents with reference beads. 50μl of EDTA uncoagulated whole blood was added to the two-tubes (CD4/CD3 reagent tube, CD8/CD3 reagent tube) using a preprogrammed electronic pipette. They were vortexed for 5 seconds and incubated in the dark at room temperature for 30 minutes. Then, 50 μl of a fixative solution was added to the tubes. The tubes were vortexed, and the non lysed stained sample was analyzed in FACSCount using the standard software.  Lines 140-148

Reviewer 2 Report

Comments and Suggestions for Authors

This manuscript tried to explore the effect of helminthiasis on host immunity, particularly in tuberculosis (TB) infection. In general, the cohort is very precious and interesting, however there are a few major concerns:

1. The authors are investigating the effect of Helminth infection. But they didn't have a group with only Helminth infection.

2. The authors only measured the total CD4, CD8 T cell frequencies. It will be great to compare different CD4 T cells subsets among the groups, because theoretically, the Mtb and Helminth will introduce different types of CD4 T cells responses.

3. The author didn't show any effects of Helminth treatment on the cell frequency. 

Comments on the Quality of English Language

The english is good to me. Maybe some minor edits or proofreading is needed.

Author Response

Response to Reviewer 2 X Comments:

Dear Madam,

Dear Sir,

Best regards

Review Report Form

Open Review

Quality of English Language

( ) I am not qualified to assess the quality of English in this paper
( ) English very difficult to understand/incomprehensible
( ) Extensive editing of English language required
( ) Moderate editing of English language required
(x) Minor editing of English language required
( ) English language fine. No issues detected

Yes

Can be improved

Must be improved

Not applicable

Does the introduction provide sufficient background and include all relevant references?

(x)

( )

( )

( )

Is the research design appropriate?

(x)

( )

( )

( )

Are the methods adequately described?

(x)

( )

( )

( )

Are the results clearly presented?

(x)

( )

( )

( )

Are the conclusions supported by the results?

(x)

( )

( )

( )

Comments and Suggestions for Authors

This manuscript tried to explore the effect of helminthiasis on host immunity, particularly in tuberculosis (TB) infection. In general, the cohort is very precious and interesting, however there are a few major concerns:

  1. The authors are investigating the effect of Helminth infection. But they didn't have a group with only Helminth infection.

Answer: We thank the reviewer for his input. The inclusion criteria for the study excluded Mtb-/Helm+ participants (see FlowChart and Table 2) and comply with the initial protocol submitted to the ethics committee. Indeed, we were interested in exploring TB-Helminthiasis interactions. We therefore compared data from the control group without infection (Mtb-/Helm-) with those from the groups with tuberculosis without helminth infection (TB+/helm-) and tuberculosis with helminth infection (TB+/Helm+). 

  1. The authors only measured the total CD4, CD8 T cell frequencies. It will be great to compare different CD4 T cells subsets among the groups, because theoretically, the Mtb and Helminth will introduce different types of CD4 T cells responses.

Answer: We thank the reviewer for his input. In the limitation as well in conclusion of the study, we stated as follow:  A thorough characterisation of the immune cells to assess their functional status was not performed for logistical reasons. This study provided a preliminary understanding of the dynamics of the immuno-haematological parameters. However, studies involving in-depth exploration of the functionality of the immune system cells are needed. Line 339-341; Lines 347-351 

  1. The author didn't show any effects of Helminth treatment on the cell frequency. 

Answer: We thank the reviewer for this input. Blood was collected in EDTA tubes from participants just once, at month zero (M0) prior to treatment initiation for both TB+/Helm- and TB+/Helm+ patients.  Due to logistical reasons, we did not take further samples at M2 and/or M6 to assess treatment efficacy. Similarly, in-depth characterization of immune cells to assess their functional status was not carried out for logistical reasons. In fact, this is a question we will be exploring further, in order to understand the role of helminths in the failure of anti-tuberculosis treatments. 

Comments on the Quality of English Language: The english is good to me. Maybe some minor edits or proofreading is needed.

Answer: We thank the reviewer for his comment. We have proofread to ensure grammatical and typological corrections.

Round 2

Reviewer 2 Report

Comments and Suggestions for Authors

After revision, the manuscript is ready for publication. Thanks!